# Laser Surgical Approach of Upper Labial Frenulum: A Systematic Review

**DOI:** 10.3390/ijerph20021302

**Published:** 2023-01-11

**Authors:** Angelo Michele Inchingolo, Giuseppina Malcangi, Irene Ferrara, Fabio Viapiano, Anna Netti, Silvio Buongiorno, Giulia Latini, Daniela Azzollini, Nicole De Leonardis, Elisabetta de Ruvo, Antonio Mancini, Biagio Rapone, Daniela Di Venere, Assunta Patano, Pasquale Avantario, Gianluca Martino Tartaglia, Felice Lorusso, Antonio Scarano, Salvatore Sauro, Maria Celeste Fatone, Ioana Roxana Bordea, Francesco Inchingolo, Alessio Danilo Inchingolo, Gianna Dipalma

**Affiliations:** 1Department of Interdisciplinary Medicine, School of Medicine, University of Bari “Aldo Moro”, 70124 Bari, Italy; 2UOC Maxillo-Facial Surgery and Dentistry, Department of Biomedical, Surgical and Dental Sciences, School of Dentistry, Fondazione IRCCS Ca’ Granda, Ospedale Maggiore Policlinico, University of Milan, 20100 Milan, Italy; 3Department of Innovative Technologies in Medicine and Dentistry, University of Chieti-Pescara, 66100 Chieti, Italy; 4Dental Biomaterials and Minimally Invasive Dentistry, Department of Dentistry, Cardenal Herrera-CEU University, CEU Universities, C/Santiago Ramón y Cajal, s/n., Alfara del Patriarca, 46115 Valencia, Spain; 5PTA Trani-ASL BT, Viale Padre Pio, 76125 Trani, Italy; 6Department of Oral Rehabilitation, Faculty of Dentistry, Iuliu Hațieganu University of Medicine and Pharmacy, 400012 Cluj-Napoca, Romania

**Keywords:** frenectomy, laser, oral surgery, upper labial frenulum

## Abstract

An abnormal and hypertrophied upper labial frenulum (ULF) can cause diastemas, gingival recession, eruption abnormalities, and the onset of carious and periodontal problems in the upper central incisors, as well as aesthetic and functional disorders of the upper lip. The goal of this investigation is to review the evidence on the surgical techniques that are currently available for treating ULF in order to identify the best approach. PubMed, Scopus, Cochrane Library, and Embase were searched for papers that matched our topic from 13 November 2012 up to 22 November 2022 using the following Boolean keywords: “frenulum” and “surgery*”. A total of eight articles were selected for the purpose of the review. ULF can be surgically treated using either traditional scalpel surgery or laser surgery. The latter is the better option due to its intra- and post-operative benefits for both the patients and the clinicians, in terms of faster healing, fewer side effects and discomfort, and greater patient compliance. However, a higher learning curve is required for this technique, especially to calibrate the appropriate power of the laser. To date, it is not possible to identify which type of laser achieves the best clinical results for the treatment of ULF.

## 1. Introduction

The medial upper labial frenulum (ULF) is a thin band or fold of mucus tissue extending from the middle of the maxillary gingiva to the center of the upper lip. Its insertion height is variable. It develops from the tecto-labial bands connecting the upper lip to the tecto-labial papilla, which will originate the incisive papilla in the third to fourth month of intrauterine life [1].

The position of the frenulum remains stable from birth until the eruption of the maxillary central incisors. Atrophy and translocation of the insertion in the apical direction are correlated with the development of the permanent teeth and the growth of the alveolar process. During primary dentition, although the frenulum insertion remains stationary, fresh bone deposits heighten the alveolar ridge. The permanent central upper incisors usually erupt with a diastema. The thrust of lateral incisors and canines (11–12 years) tends to minimize the diastema and bring the incisors closer together [1,2].

The frenulum is mostly made of collagen, although it also has elastic and lax reticular fibers. The lining is distinguished by stratified compound pavimentous epithelium.

It is unknown if there is muscle tissue in the frenulum, however, it is now believed that muscle tissue is not an essential component of the frenulum [3]. Due to the lack of muscle fibers and developmental behavior, the frenulum does not appear to affect tooth emergence, gingival morphology, or the mobility of mucosal structures. At this time, an interincisal diastema is not thought to be determined by the persistence of frenulum insertion at the level of the papilla [1,4].

Different classifications for ULF have been proposed by authors, based on the place of implantation and site of insertion. Placek in 1974 classified the different forms of frenulum according to the place of implantation, distinguishing (a) mucosal attachment frenulum when the frenulum is inserted into mucosa; (b) gingival attachment frenulum, when it is inserted into adherent gingiva; (c) papillary-attached frenulum, when it inserts into the palatine papilla; and (d) penetrating papillary attachment frenulum, when the frenulum fibers pass through the interincisive papilla and insert into palatine adherent gingival [5,6].

In 1977, Popovich et al. classified the frenulum according to the site of insertion. In this case, the frenulum is divided into two primary morphotypes, thick and thin, which can be distinguished into in: high (alveolar mucosa), medium (adherent gingiva), and low (marginal gingiva) [7].

Likewise, Rui et al. proposed a classification of frenulum considering site insertion: (a) Type I: alveolar mucosa; (b) Type II: gingival insertion; (c) Type III: interdental papilla; and (d) Type IV: transpapillar [8].

Generally, the type of insertion, the width of the frenulum, the mechanical action on the gingival margin, and the difficulty in performing dental care procedures have all been regarded as criteria to identify a normal frenulum from a defective one without ever arriving at a uniformity of evaluation.

A characteristic of an abnormal frenulum, known as “pull syndrome,” is the ischemization of the palatine papilla and mesial gingival borders of the upper central incisors after traction of the top lip (Figure 1).

Others experience the taut, fibrous, papillary, or transpapillary insertion interincisional frenulum more frequently, which has a broad triangular upper insertion known as the tecto-labial frenulum. This kind of frenulum is regarded as a genuine dysontogenic manifestation of fetal frenulum persistence after birth [5]. The surgical approaches that are currently in use are: (a) frenectomy or excision, which involves complete removal of the frenulum; (b) frenulotomy, which involves partial removal of the frenulum (Figure 2); (c) apical repositioning of the frenulum [9,10,11].

The types of surgical techniques that are currently available are conventional scalpel surgery and laser surgery.

### 1.1. Conventional Surgery

The surgical techniques are: (a) classical frenectomy/V-shaped Archer/diamond-shaped incision; (b) Z-plasty incision; (c) extension of the vestibular groove; and (d) frenectomy according to Miller’s technique.

The conventional cold-blade method has all the side effects of classical surgery, such as tissue bleeding with the need for suturing and possible bacterial overinfection of the surgical site [12,13].

Classical frenectomy/V-shaped Archer/diamond-shaped incision: devised by Archer (1961) and Kruger (1964). The cold blade procedure is the traditional technique and involves a V-shaped incision along the edges of the frenulum, removal of the coronal insertion of the frenulum with subsequent apical repositioning. The periosteal fibres are then cut horizontally and vertically, and the wound is closed with sutures. This technique involves more side effects because it produces the development of scar tissue with possible periodontal repercussions until the disappearance of the interincisive papilla (Figure 3).

Z-plasty incision: uses a Z-shaped incision to create two triangular flaps of equal size that are then transposed and sutured. The stitches are removed 5–7 days after surgery. It has minor scar outcomes and is indicated in cases of the deep fornix, with wide, hypertrophic frenules and low insertion. It determines the partial removal of frenulum tissue that is associated with the deepening of the fornix. It may cause recurrence (Figure 4).

Extension of the vestibular groove: mucosa is incised and repositioned more apically, with suturing to the periosteum and healing by second intention. It has a high relief rate.

Frenectomy according to Miller’s technique: the free gingival pedicle is placed laterally in order to reduce scar tissue formation. Although this incurs a more complex procedure for the operator, it has an aesthetic advantage and it is particularly indicated in cases of a gingival smile.

Electrosurgery: recommended in patients with coagulation problems. It also provides effective hemostasis in patients with poor compliance [14].

### 1.2. Laser Surgery

Nowadays, different types of lasers are commonly used in everyday practice. Each one produces a different wavelength of light, which is defined by the active medium (solid, liquid, or gas) [15].

Lasers that are actually used in oral surgery are: Potassium titanyl phosphate laser, or KTP (532 nm); diode laser (810–930 nm); neodymium-doped yttrium aluminum garnet laser, or Nd:YAG (1064 nm); erbium family lasers (2780–2940 nm), which includes the erbium-doped yttrium aluminum garnet laser, or Er:YAG; the erbium chromium yttrium scandium gallium-garnet laser, or Er,Cr:YSGG; and a carbon dioxide laser, or CO_2_ (10,600 nm) [15,16] (Figure 5).

The fundamental distinction among them is the capacity to operate strictly on soft or hard tissues or both [17,18]. When just soft tissue surgery is required, the choice of the clinician must fall upon KTP, Diode, Nd:YAG or CO_2_ lasers [19,20]. Instead, the erbium family lasers (Er;Cr:YSGG or Er:YAG lasers) may be required when both soft and hard tissue surgery is necessary [20,21,22] (Figure 6).

Several authors have highlighted the benefits of lasers over other types of traditional dental techniques, namely increased coagulation and hemostasis, which results in a dry surgical area for greater vision and a significantly diminished requirement for suturing (the erbium family lasers are an exception to this rule as they offer only a little haemostasis); decrease in bacteremia (tissue temperature changes brought on by laser surgery are efficient at reducing bacteria); accelerated healing process (laser beam can promote healing via photo-biomodulation); and reduced post-surgical pain [17,20,23,24,25].

Furthermore, the diode laser has become highly popular in dental practice given its small size, low cost and easy application for minor soft tissue surgery, for its properties of straightforward tissue incision, and its coagulation and post-surgery benefits [26,27] (Figure 7).

Regardless of the surgical technique that is used, labial frenulums are anatomical structures with high biological memory, so surgery should include not only the removal of the frenulum, but also that of its deep periosteal insertions up to the muco-gingival line, in order to prevent a recurrence. Fibrotomy of the circular and transseptal fibers should be prolonged up to the level of the palatal bone in cases of Type IV maxillary labial frenulum with palatine insertion [9,28,29].

There are already numerous articles in the literature dealing with the lingual frenulum and its resolution, instead, although ULF is the most frequent type of frenulum and has the greatest impact on orthodontic treatments, there is little evidence about its surgical treatment. Herein, the purpose of this study is to review the surgical techniques that are currently available for the treatment of ULF, analyzing the advantages and disadvantages of each approach.

## 2. Materials and Methods

### 2.1. Protocol and Registration

The Preferred Reporting Items for Systematic Reviews and Meta-Analyses (PRISMA) guidelines were used in this systematic review [30]. The review protocol was registered at PROSPERO under the unique number 381024.

### 2.2. Data Sources and Search Strategy

The qualifying criteria were developed using the PICOS (population, intervention, comparison, outcomes, and study design) framework. Pubmed, Cochrane, Scopus, and Embase databases were searched from 13 November 2012 up to 13 November 2022, using the keywords “frenulum” and “surgery”. The authors checked the titles and complete texts of any papers that might be relevant. Table 1 summarizes the search approach in detail (Table 1).

### 2.3. Inclusion and Exclusion Criteria

Articles that met the following criteria were included: (1) studies only on humans; (2) open access studies; (3) randomized trials, retrospective and observational studies; and (4) studies dealing with surgical approach of ULF and comparing the different techniques.

The exclusion criteria were: reviews, letters to the editors; animal models or dry skulls studies; and craniofacial syndromes, or cleft lip and palate.

### 2.4. Data Collection

The study data were selected by analyzing the study design, sample size, control group, age of intervention, type of surgery, follow-up, and outcome (Table 2).

## 3. Results

### Study Selection and Included Study Characteristics

The electronic database search identified a total of 334 studies. After duplicate removal, 274 studies underwent title and abstract screening. In total, 274 papers were not selected after the abstract screening, and 56 articles were chosen for the eligibility assessment. Subsequently, 48 papers were eliminated after the full-text evaluation: 42 off-topic, 5 wrong setting, and 1 with no outcome of interest. Finally, 8 articles were selected for the systematic review. The selection process is summarized in Figure 8.

## 4. Discussion

The presence of an abnormal and hypertrophied interdental ULF could cause diastemas, gingival recession, eruption abnormalities, and the onset of carious and periodontal problems of the upper central incisors, as well as aesthetic and functional disorders of the upper lip [8,33]. The surgical approach to solving the problem involves the use of a conventional scalpel [31,36,37], electrosurgery, or the use of different types of high-intensity laser devices [8,31,32,33,34,35,36,37]. Although electrosurgery is a well-known surgical technique for ULF [14], in this review only studies about traditional surgery and laser surgery have passed the screening phase. The surgical techniques that are used have always been studied and compared, and although no study emphasizes which technique is the most effective, the literature is unanimous that conventional surgery unless combined with a Z-plasty is contraindicated in the presence of a short lip and fornix, as the scarring outcome would worsen the initial clinical situation [32,38].

In the case of conventional scalpel surgery, suturing is always necessary; the electric scalpel has great hemostatic capabilities, but it causes cicatricial outcomes and often postoperative pain; the laser irradiation, on the other hand, under the correct parameters does not penetrate deeply into the tissues, but draws precise, non-bleeding surgical margins, consequently improving the visibility of the surgical site and the comfort and compliance of the patient [8,39].

Studies comparing traditional surgical techniques vs. diode laser surgery highlight the superiority of the laser surgery technique in terms of hemostasis, surgical time, pain, edema, post-surgical inflammation, and wound healing time [35,36,37].

In a clinical retrospective study, Komori et al. analyzed the use of CO_2_ laser with a wavelength of 10.6 μm on six pediatric patients, of which two presented Type II frenulum and four presented Type III frenulum according to Rui’s classification. Suturing was necessary for only one patient out of the six that were treated, as the high-frequency laser irradiation demonstrates excellent hemostatic activity, as well as precise resection and vaporization of the soft tissues of the oral cavity. None of the patients complained of postoperative problems such as pain, difficulty in chewing, hemorrhage, and recurrence with fiber reattachment. The surgical time is short and the technique is simpler than traditional surgery [8,40,41,42,43].

Ling et al. compared the conventional surgical approach with a frenectomy procedure that uses two successive pulses of the Er:YAG and Nd:YAG lasers. The authors demonstrate that the laser surgery technique is more comfortable and less painful for the patient both intra-operatively and post-operatively, and chair time is reduced as there is no need to suture and then remove the suture, so there is an increase in patient cooperation. The patient does not even need to take antibiotic therapy due to the laser irradiation’s ability to sterilize and hemostasis the area. On the other hand, the conventional surgical technique may result in longer healing times if the patient does not observe strict oral hygiene leading to plaque accumulation on the sutures, but it has the advantage of having a shorter learning curve for the operator and being less expensive [31,44].

In a prospective study of 50 patients, Pie-Sanchez et al. compared the efficiency of two types of laser devices: CO_2_ and Er,Cr:YSGG concluding that the CO_2_ laser allows the operator to perform surgery in a shorter time and with a bloodless surgical field, but the Er,Cr:YSGG laser gives faster wound healing [32,45,46].

The versatility of the laser surgery technique was illustrated in Rozo’s case study. The 16-year-old patient received frenolectomy and anterior gingivoplasty in the same procedure, the first for functional reasons, and the second for artistic ones. Laser intervention may be utilized for two treatments at once, cutting down on visits and discomfort for patients [35].

Sfasciotti et. al. analyzed frenectomy treatment in pediatric patients that were at risk of developing gingival recessions aggravated by the presence of the frenulum. The study compared diode vs. CO_2_ laser appliances and concluded that the former is a better performer in terms of pain management and wound healing speed, but the latter gives better bleeding control [33,47,47,48,49].

In their work, Onur et al. analyzed pain perception and wound healing time in patients that were treated with Er,Cr: YSGG vs. diode laser surgery, with no statistically significant differences between the two groups with regards to safety and efficiency [3,34,50].

When employing lasers, caution must be taken to calibrate the appropriate power and to safeguard the operator’s and patient’s eyes using a particular equipment. Excessive power can cause damage to the bone surface [8,32]. Diode, Nd: YAG, and CO_2_ lasers in particular overheat nearby tissues, and any fiber interaction with the periosteum must be avoided since this might result in localized, reversible tissue necrosis and pain for the patient [32,36].

The main limitation of this study is that although studies examined analyze the same clinical parameters, they use different classifications and compare one type of laser with the traditional surgical technique or only two types of laser appliances with each other. As a result, it is impossible to make an exact comparison between the traditional surgical technique and the different types of laser techniques.

## 5. Conclusions

Diastemas, gingival recession, eruption abnormalities, carious and periodontal lesions of the upper central incisors, as well as aesthetic and functional lip disorders, can all be caused by an abnormal median ULF.

This systematic review includes studies that examine the surgical techniques that are available for treatment, including traditional surgery and laser surgery.

Although traditional scalpel surgery has a shorter learning curve and lower costs, it has several drawbacks, including longer operating times, bleeding, decreased visibility of the surgical field, the need for tissue suturing, longer healing times, and the risk of plaque build-up on the sutures. In contrast, the laser surgery approach allows for treatment with significant reductions in intra-operative and post-operative discomfort for both the operator and the patient. It enables clean, bleed-free margins that improve surgical site visibility and do not require suturing. Based on the examination of the chosen articles, it appears that the laser surgery technique appears to be better in terms of haemostasis, surgical time, pain, edema, post-surgical inflammation, and healing time when compared with conventional surgery.

Despite this, it has not been possible to determine which type of laser source has the best clinical results. Consistent randomized controlled trials are necessary to compare the benefits and drawbacks of the various surgical laser devices currently available.

## Figures and Tables

**Figure 1 ijerph-20-01302-f001:**
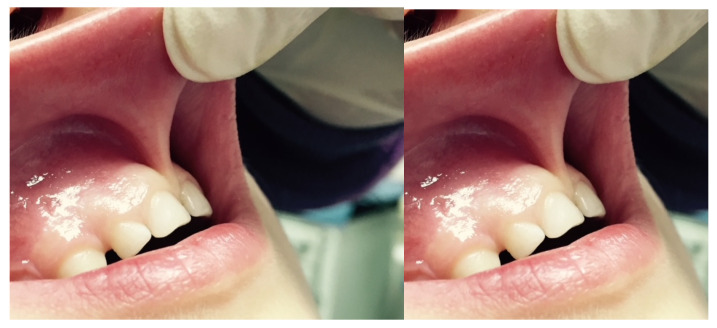
Example of the hypertonic ULF.

**Figure 2 ijerph-20-01302-f002:**
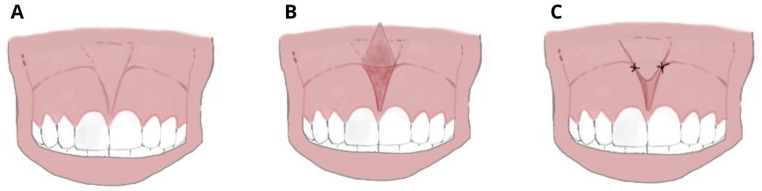
Frenulotomy: (**A**) Hypertrophic superior median frenulum with insertion on the interincisive papilla; (**B**) partial thickness incision disengages frenulum and muscle fibers and does not involve the periosteum; and (**C**) suture and apical repositioning on the labial side.

**Figure 3 ijerph-20-01302-f003:**
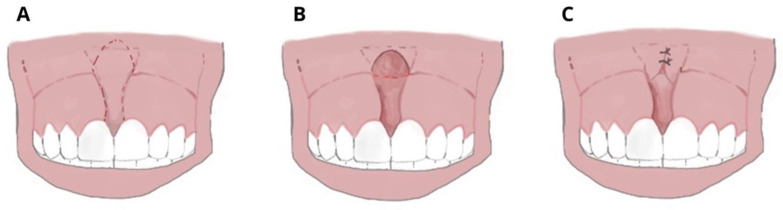
Frenectomy with traditional and V-shaped surgery: (**A**) Transpapillary frenulum; (**B**) two paramedian vertical incisions are drawn parallel to the sides of the frenulum up to the retroincisive papilla, where they connect with a horizontal cut; the frenulum is raised and the periosteum is exposed; (**C**) the margins are sutured along the midline, not involving the adherent gingiva; and periodontal wrap and healing by second intention.

**Figure 4 ijerph-20-01302-f004:**
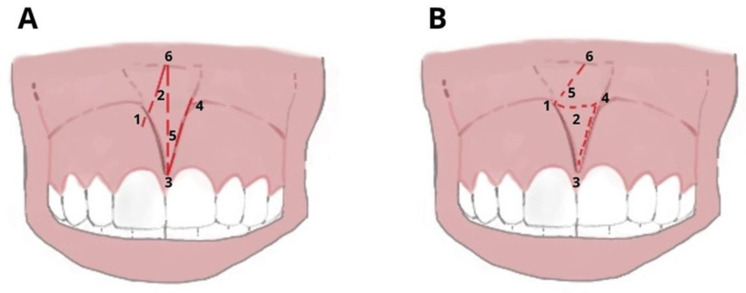
Z-Plasty: (**A**) Vertical incision in the center of the frenulum, with releases parallel to 60° to form a Z and (**B**) transposition of the mucous flaps of triangular shape and suture: the orientation of the fibers that were vertical rotates and becomes horizontal. (**1**–**6**) The numbers refer to the chronology of execution of the incision.

**Figure 5 ijerph-20-01302-f005:**
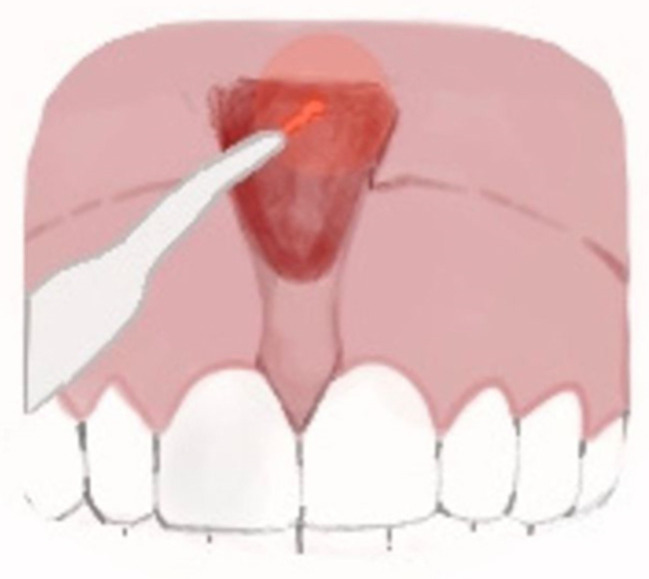
Laser frenectomy.

**Figure 6 ijerph-20-01302-f006:**
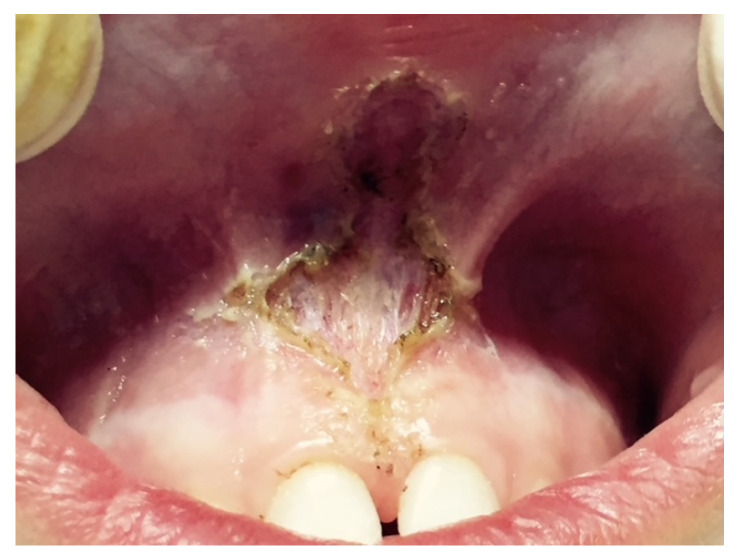
ULF laser surgery.

**Figure 7 ijerph-20-01302-f007:**
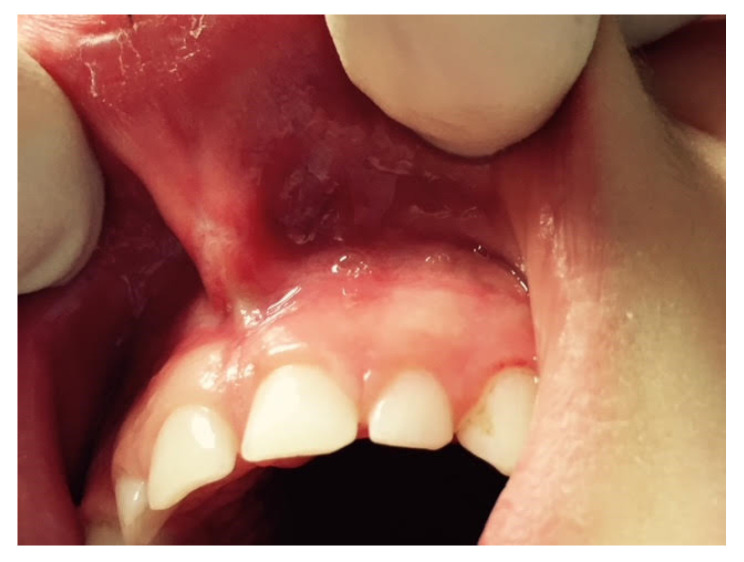
Labial frenulum treated by laser frenectomy after healing.

**Figure 8 ijerph-20-01302-f008:**
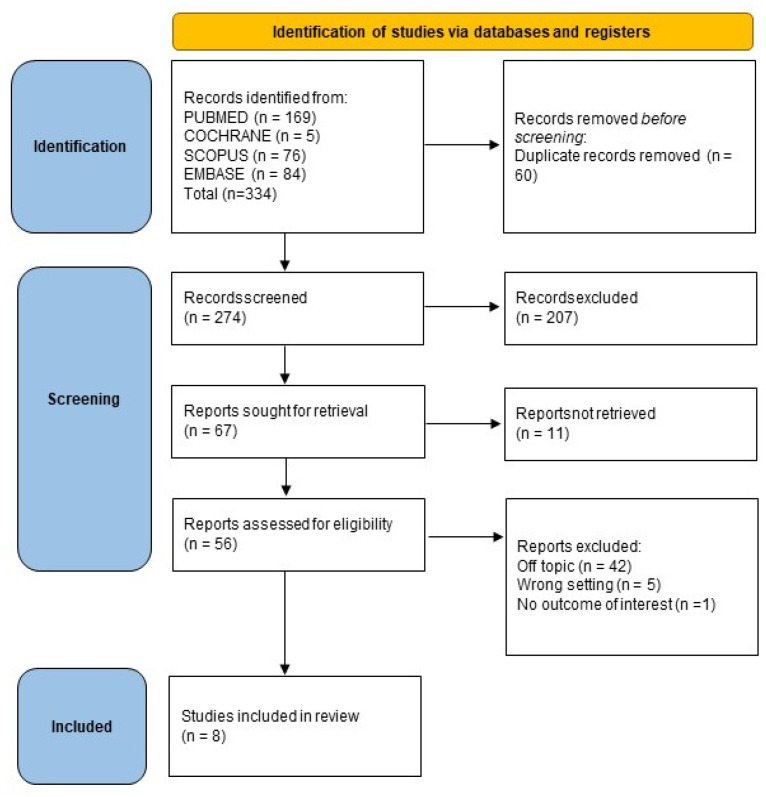
Literature search preferred reporting items for systematic reviews and meta-analyses (PRISMA) flow diagram.

**Table 1 ijerph-20-01302-t001:** Database search indicator.

Articles screening strategy	KEYWORDS: A: “frenulum”; B: “surgery”
	Boolean Indicators: (“A” AND “B”)
	Timespan: from 13 November 2012 to 13 November 2022
	Electronic Database: Pubmed, Cochrane, Scopus and Embase

**Table 2 ijerph-20-01302-t002:** Studies included characteristics [CO_2_: carbon dioxide laser, DB-RCT: double bind-randomized controlled trial, Er,Cr:YSGG: erbium chromium yttrium scandium gallium-garnet laser, Er:YAG: erbium-doped yttrium aluminum garnet laser KTP: potassium titanyl phosphate laser. Nd:YAG: neodymium-doped yttrium aluminum garnet laser, RCT: randomized controlled trial].

Author/ Year	Study Design	Sample Size	Average Age at Intervention (Years)	Type of Surgery	Follow-Up
Komori S. et al. 2017 [8]	Retrospective	21 total laser surgery 15 lingual 6 labial	6.0 Lingual: 5.2 Labial: 8.2	CO_2_ laser with a continuous wave at 2 to 5 W for ~60 s	4.6 months Re-adhesion in 1 patient of the lingual frenulum
Xie L. et al. 2021 [31]	DB-RCT	34 patients 17 laser surgery 17 scalpel surgery	5–10	Laser surgery: First, Er:YAG laser with wavelength 2940 nm short pulse (SP); energy: 60 MJ; Frequency: 30 Hz; power: 1.80 W. Subsequently, Nd:YAG laser, were set as pulse width: very long pulse; frequency: 20 Hz; power: 4.00 W	1 month Wound healing and no re-adhesion in every group Scar in 1 case of scalpel surgical technique
				Scalpel surgery: used sterile scalpels, #11 and 4–0 absorbable suture	
Pie-Sanchez J. et al. 2012 [32]	RCT	50 total patients 25 CO_2_ laser surgery 25 Er,Cr:YSGG laser surgery	11.3	CO_2_ laser with 10,600 nm wavelength used in the focused continuous wave mode, a power rating of 5 W	4 months No difference in post-operatory Intraoperative, bloodless field and faster operation time in CO_2_ laser group Faster wound healing in Er,Cr: YSGG laser group
				Er,Cr:YSGG laser 2780 nm wavelength, a pulse duration between 140 and 200 µs, and a 20 Hz frequency. Power settings: 1.5 W with 12% water and 8% air	
Sfasciotti G.L. et al. 2020 [33]	DB-RCT	26 total patients 13 Diode laser surgery 13 CO_2_ laser surgery	9	The diode laser 980 nm wavelength with a continuous modality of pulse and set at 2.5 W	14 days Better biological results and fewer intraoperative errors in Diode laser group
				The CO_2_ laser with 10.600 nm in a super pulse wave modality set at 4.5 W	
Onur S.G et al., 2020 [34]	Retrospective	22 total patients 11 Er,Cr:YSGG laser surgery 11 diode laser surgery	8–13	2780 nm Er,Cr:YSGG laser: power 2.75 W; frequency, 50 Hz; pulse duration, 60 ls; pulse energy, 55 mJ/pulse; energy density per pulse, 22 J/cm^2^; 20% air, 40% water	2 weeks Better wound healing in the Er,Cr:YSGG laser group
				940 nm diode laser operated at a power of 1.5 W incontinuous wave mode	
Pulido Rozo M.A. et al., 2015 [35]	Case report	1	16	High-intensity laser not specified	15 days Post-operative edema resolved and good healing
Junior et al., 2015 [36]	Prospective	40 total patients 22 scalpel surgery 18 Nd:YAG laser surgery	20.9 ± 10.3	Scalpel blade no. 15 and simple suture with silk thread 4–0	Reduced surgery time and no sutures in Nd:YAG laser group
				Nd:YAG laser (λ = 1064 nm) with the following parameters: 40 J of energy, 40 Hz frequency, 4 W of power, for 10 s (power density = 5 W/cm^2^ and energy density = 50 J/cm^2^), and a short pulse width	
Pinheiro A. et al., 2018 [37]	Case series	1 scalpel surgery 1 diode laser surgery	28	An incision with scalpel number 15 and suture with nylon thread 5.0	7 days More time-effective and less intra- and post-operative bleeding in laser surgery Post-operative drug intake, discomfort, and edema in scalpel surgery
				Diode laser infrared wavelength 808 nm infrared 808 nm, power of 2 W, the energy of 120 J, pulsed mode pulsed mode and 20 pps repetition frequency	

## Data Availability

No applicable.

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
