# Peer review of "Laser Surgical Approach of Upper Labial Frenulum: A Systematic Review"

_ijerph, 2023, doi:10.3390/ijerph20021302_

Round 1

Reviewer 1 Report

1. I remain unconvinced that a normal upper lip frenulum causes any practical of significant cosmetic issues in children. In 30 years of Paediatric Surgical care I can recall only one of two patients in whom the frenulum extended significantly onto the marginal gingiva. I have seen several children in whom the frenulum has been torn following facial trauma. 

2. In relation to the surgical approach, I have used needle-point diathermy on a low setting. This produces excellent results at low cost, is quick, easy to use and requires no special equipment, training or Laser precautions. It is disappointing that the authors rather dismiss this technique due to a lack of evidence.

Reviewer 2 Report

dear authors, the topic is interesting, but i feel it requires some modifications in all the sectors. Major language and spelling corrections required. There is no continuity in the introduction. All the included studies have lasers commonly used, so why the title in general?

Reviewer 3 Report

Dear authors,

Thank you very much for submitting your paper to the prestigious IJESPH.

I hope that my remarks will be useful in order to increase the quality of the paper.

1.     First of all the number of authors is extremely high. 23 authors for a paper? Please kindly check the authorship criteria.

2.     3 authors with euqual contribution as the first author, 2 authors with equal contribution as the last author and 2 corresponding authors? I mihgt think about fabrication of authorship.

3.     Why is „scalpel surgery” a keyword?

4.     Placek classification should be included as it is very recent.

5.     Line 119 – Please replace „traditional” with „conventional”

6.     Line 161 – Please include the laser assisted surgical protocol taking into account the type of laser used and the energy.

7.     Line 284- This should belong to „Materials and methods”.

8.     Line 396- The stament is misleading

9.     Line 397- 399- There is no evidence in your paper to support that.

10.  You have a paper of 399 lines and 23 authors. When I divided it it revelead that every author wrote 17 lines. Please try to solve this issue.

Kind regards!

Reviewer 4 Report

Dear Authors

This systematic Review is an interesting paper. Still there are some points that need to be rectified, in order to improve the content and make it more accurate for the readership.

Discussion Section:

In this section, laser should be mentioned as laser irradiation either laser device or laser surgery (depending on the sentence).

E.g. Line 331 "the laser, on the other hand, does not penetrate deeply into the tissues, but 331 draws precise, non-bleeding surgical margins, consequently improving the visibility of 332 the surgical site and the comfort and compliance of the patient [7]. "

It is Laser irradiation.

Also, In the same sentence:

Lasers do penetrate deep into tissues, especially Nd:YAG

[Celeste L. Saucedo, Emily C. Courtois, Zachary S. Wade, Meghan N. Kelley, Nusha Kheradbin, Douglas W. Barrett, F. Gonzalez-Lima. Transcranial laser stimulation: Mitochondrial and cerebrovascular effects in younger and older healthy adults (Brain Stimulation, Volume 14, Issue 2, 2021, Pages 440-449)].

Thus it should be rephrased, and include the important words "under the correct parameters".

References Section: 

Reference No 37 is written incorrectly, please rectify.
References No38 to the end, have no connection to this article - please remove them (from the main body of Discussion Section and the Reference List), or replace them with appropriate ones.

Looking forward to receiving a revised manuscript

Round 2

Reviewer 2 Report

no suggestions